# Self-Care Ability and Life Quality of Cured Leprosy Patients: The Mediating Effects of Social Support

**DOI:** 10.3390/healthcare11233059

**Published:** 2023-11-28

**Authors:** Li Xu, Guangjie Jin, Xiang Li, Yuting Shao, Yunhui Li, Lianhua Zhang

**Affiliations:** 1Key Laboratory of Environmental Medicine Engineering of Ministry of Education, Southeast University, Nanjing 210003, China; 220214010@seu.edu.cn (L.X.); 23021847@seu.edu.cn (X.L.); 220213976@seu.edu.cn (Y.S.); 2Department of Chronic Infectious Disease Control and Prevention, Jiangsu Provincial Center for Disease Control and Prevention, Nanjing 210009, China; jingj@jscdc.cn

**Keywords:** leprosy, cured leprosy patients, social support, self-care ability, quality of life

## Abstract

Objectives: The study explores the relationship between social support, self-care ability, and life quality of cured leprosy patients (CLPs), aiming to develop strategies to enhance their overall well-being. Methods: From July to December 2021, we investigated the social support, self-care ability, and life quality of CLPs through three scales and analyzed the correlation between them. In addition, structural estimation modeling (SEM) was employed to analyze their correlation. Results: A total of 9245 CLPs were recruited, with a male-to-female ratio of 2.19:1, and 94.04% of cured patients was 60 years or above, with predominantly home-cured patients. The scores of WHOQOL-BREF, SSRS, and ESCA were (51.39 ± 9.89), (31.87 ± 8.76), and (100.95 ± 19.75), respectively. The results indicate a poorer quality of life and social support for CLPs compared to the general population in China. Furthermore, the home group had higher scores on these scales than the leprosarium group. The correlation analysis showed significant interactions between life quality, social support, self-care ability, and various domains (*p* < 0.05). SEM results revealed that the direct effect of self-care ability on life quality was 0.13, and the indirect effect on quality of life through social support was 0.08. The mediating effect of social support accounted for 22.86% of the total effect in the home group. In the leprosarium group, the effect of self-care ability on quality of life was 0.14. Conclusions: Most CLPs in Jiangsu Province are concentrated in the central region, with a high disease burden. We found that CLPs have a poorer life quality than the general population, with the leprosarium group being worse than the home group. The government and society should pay more attention to and support these cured patients.

## 1. Introduction

Leprosy is a chronic infectious disease caused by *M. leprae* or *M. lepromatosis*, which mainly affects the human skin, mucous membranes, and peripheral nerves [1,2,3]. A delayed diagnosis of leprosy is more common and causes irreversible damage to the peripheral nerves of the patients, eventually leading to disfigurement and disability [4]. According to the World Health Organization (WHO), leprosy is still spreading in 143 countries, with a total of 174,087 new cases detected in 2022 [5]. The burden of leprosy for patients remains high. For instance, in a cohort study conducted by Richardus JH et al. [6], they found that about 6.6% of leprosy patients developed a new nerve function impairment after registration. Disability-adjusted life years (DALYs) measure the loss of health due to fatal and non-fatal disease burdens, and an examination of levels and trends in DALYs facilitates rapid comparisons across diseases and injuries. Stolk et al. [7] found that DALYs increased from 36,000 to 41,000 in the period of 1999–2010. Meanwhile, lower middle income regions accounted for about 74% of all DALYs from leprosy.

Although the annual detection rate (NCDR) of leprosy in China has declined, the rate of deformities in new cases is still at a high level, and the number of years of healthy life lost due to deformities is high. The Jiangsu Province used to have one of the most considerable rates in China, with a total of 56,176 leprosy cases detected and registered from 1949 to 2020. A previous study [8] showed that, during the period of 2005–2020, there was a 31.68% of newly discovered G2D patients and it was significantly higher than in other endemic areas of China, such as Yunnan, Sichuan, and Guizhou. However, leprosy burden is often misinterpreted as the eradication or the absence of major complications in Jiangsu. The disease may also progress gradually over time, which means that quantifying the burden at a given time is not representative over the entire course of leprosy. Experts has pointed out that the chronic consequences of leprosy, such as the negative impact on social participation and mental health, are not taken into account. In addition, discrimination is an important cause of delayed diagnosis and promotes the widespread spread of *M. leprae* within the family and community contexts. Even more seriously, discrimination against leprosy patients seriously affects the quality of life of patients in China, especially among CLPs.

In this study, CLPs were those who received anti-leprosy treatment and met the criteria for cure (Chinese Standard). The policy stipulated that individuals who had been cured of leprosy could opt to remain in the leprosarium if they lacked a home to return to after their recovery. Even though they are cured, they are ostracized by the community, and such discrimination could be perpetrated by neighbors, family members, and medical staff [9,10,11]. This discrimination reduces their employment opportunities, and although the government provides subsidies to them, it is only provided to cured patients in the leprosarium. Such economic deprivation, physical disability, and stigmatization of the disease continue to have a negative impact on their life quality. Therefore, it is urgent to investigate this vulnerable group, and improve the quality of their life. However, most studies in China have focused on the survival status of CLPs, and few studies have focused on their overall well-being. This situation is even more severe in Jiangsu Province, where the number of CLPs is significantly higher than in other provinces. Therefore, it is necessary to investigate CLPs to know the current status of their quality of life, social support, self-care ability, and the association between them, aiming to provide recommendations to improve their quality of life.

## 2. Methods

### 2.1. Research Context

Prior to the promotion of the use of Multidrug Therapy (MDT) programs, the Government of China segregated leprosy patients for treatment in leprosaria. The leprosarium consists of persons affected leprosy, CLPs, some of the patients’ family members, medical staff, and nursing staff. After the promotion of MDT, new cases can freely choose between isolation at home or in a leprosarium. Therefore, these are the only two types of residences available to CLPs in Jiangsu Province. Although they live in different locations, the care received by CLPs is generally the same, including primary care (help with bathing, changing clothes, cutting nails, etc.), living care (hygiene, dietary care, excretory care, etc.) rehabilitation care (functional training, social adaptation training, activity therapy, etc.), and psychological support (active listening, reassurance, guidance, encouragement, etc.).

### 2.2. Study Population

The cross-sectional study was conducted in Jiangsu Province from July 2021 to December 2021. A census was used to survey CLPs at home and those in leprosaria We checked the China Leprosy Management Information System (LEPMIS) to ensure no patients were missed. This study did not include those who were uncooperative, confused, or unable to complete the survey. The Chinese standard of CLPs is: complete disappearance of skin lesions, no pressure pain on nerve trunks, no leprosy reaction within one year, being positive for 12 consecutive months, no leprosy lesions on histopathology, and negative antacid staining.

### 2.3. Recruitment Process

The participant recruitment procedure was conducted as follows. (1) All leprosy patients were registered in the Leprosy Control Information System (LEMPIS) before the survey. Survey respondents were identified based on this system and in conjunction with the historical registry. (2) Local medical staff were uniformly trained to survey and contact CLPs. (3) Before the survey, CLPs were informed of the purpose of the study and confidentiality terms, and were asked if they were willing to participate in this study. (4) CLPs were surveyed by the medical staff in a question-and-answer format after signing an informed consent form.

### 2.4. Study Tools

The survey instruments mainly included the basic information questionnaire, WHO Quality of Life Brief Scale (WHOQOL-BREF), Social Support Rating Scale (SSRS), and Exercise of Self-care Agency Scale (ESCA). The basic information questionnaire was self-designed based on the relevant literature, which included age, gender, occupation, marriage, deformity, and disability. WHO developed the WHOQOL-BREF, which was mainly used to assess the quality of life of cured leprosy patients in our study [12]. It contained 26 entries and set 4 dimensions of physical, psychological, social relationships, and environment. SSRS [13] includes three measurements: objective, subjective support, and utilization of social support. It was used to assess the perception of and satisfaction with social support among CLPs. In previous studies [14], the scale showed a good reliability and validity. The ESCA was compiled by Kearney and Fleischer in 1979 and was translated into Chinese by Taiwanese scholars Wang et al. [15] in 2000. It showed a good reliability and validity in a sample of Taiwanese women. In this study, it was mainly used to measure the self-care ability of CLPs with high validity and reliability [16,17]. The scale consists of 43 items and four dimensions: self-concept, self-care responsibility, self-care skills, and health literacy.

### 2.5. Data Collection Process

A face-to-face questionnaire was conducted, taking into account the fact that CLPs were older, less educated, and physically disabled. Training was standardized for all investigators, and qualified investigators participated in this research. Before distributing the questionnaires, the purpose of the survey, the principle of confidentiality, and the precautions for completing the questionnaires were explained to the CLPs. The questionnaire was distributed after obtaining the informed consent of the person concerned, and those who could not fill in the questionnaire were assisted by the investigators. At the end of the survey, the questionnaire content was entered twice to establish a database.

### 2.6. Survey Flow Chart

In summary, this study comprised four key stages: Purpose, Pre-survey, Formal survey, and Data Collection and Analysis. Initially, in the Purpose stage, our focus was on measuring the disease burden of leprosy. Subsequently, during the Pre-survey phase, we meticulously devised the study protocol, coordinated active participation across regions, and conducted comprehensive training for the local medical staff. A preliminary survey involving a small group was executed to assess the study’s feasibility. Moving on to the Formal survey stage, the investigators engaged in face-to-face surveys with the CLPs. The supervision of the investigation process included informing the investigators of the management system of the investigation, such as the investigation team management methods, investigation guidance and supervision measures, and investigation summary and communication system as well as knowing the work of the investigators and solving the problems they encounter in a timely manner. In the Data Collection and Analysis phase, we established a database to systematically analyze the collected data, examining scores and interrelationships across various scales. All the research processes are shown in Figure 1.

### 2.7. Statistical Methods

All the data were entered into EpiData3.1 (Association, Odense, Denmark) to establish a parallel double-track database. SPSS 26.0 (IBM Corporation, Armonk, NY, USA) was applied for preliminary statistical analysis. Descriptive statistical analysis included the frequency and percentage of classified data and the continuous data’s mean and standard deviation (SD). *T*-tests and analysis of variance were used to compare scale scores among different groups. Spearman’s correlation analysis was used to evaluate the bivariate correlations between quality of life, self-care, and social support among the CLPs. Graphs in the study were plotted using the Prisma 9.5.0 with a 2021 origin; the flowcharts were plotted by BioRender (accessed on 20 November 2023), while the maps were drawn by Dysprosium Charts (accessed on 9 October 2023). We implemented structural equation modeling (SEM) using the AMOS 24.0 (IBM Corporation, Armonk, NY, USA) to examine further the pathway relationship between SSRS, ESCA, and QOL, as well as the potential mediating effect of SSRS between ESCA and QOL. All comparisons were two-tailed, and *p* < 0.05 was considered statistically significant.

### 2.8. Ethics Approval

In accordance with institutional guidelines and national laws, this study did not involve human clinical trials or animal experiments. It was exempt from the ethical review process by the Ethical Review Committee of the Jiangsu Provincial Center for Disease Control and Prevention. All subjects provided a written informed consent in accordance with the Declaration of Helsinki. The subjects were ensured confidentiality and anonymity. All participation was voluntary. 

## 3. Results

### 3.1. Demographic Characteristics

A total of 10,082 questionnaires were collected, of which 9245 were valid, with an adequate participation rate of 91.7%. These patients were mostly concentrated in the central part of Jiangsu Province (Figure 2), 6345 of them were male, and 2900 were female, with a male to female ratio of 2.19:1. The average age of the CLPs was (74.13 ± 9.04) years, with the maximum being 102 years and the minimum being 28 years, and 94.04% of them were aged 60 years or above. The type of residence was divided into two categories, mainly including 8685 in the home group and 560 in the leprosarium group. 

By comparing the essential characteristics of the home group and the leprosarium group, it was found that there was a statistically significant difference in the gender composition of the two groups (*p* < 0.05). In addition, there were statistical differences between the two groups regarding marital status, education level, occupation, life and work capacity, the presence of deformities, and living expenses and insurance (*p* < 0.001). However, there was no statistical difference between the two groups regarding age and account (the household registration system of China). For details, see Table 1.

### 3.2. Scores on the CLP Scale

The Cronbach′s α coefficients of the scales WHOQOL-BREF, SSRS, and ESCA used in this study were 0.914, 0.720, and 0.816, respectively, all of which were higher than 0.70. The results show that the scales we used have a high reliability and validity. Table 2 provides further details.

The scores of quality of life, social support, and self-care ability of the CLPs were (51.24 ± 9.85), (31.71 ± 8.73), and (100.82 ± 19.89), respectively. The study results show that leprosy patients have a poorer quality of life and social support after cure compared to the general population in China (Table 2). The closer to the North of Jiangsu Province, the worse the quality of life of CLPs. Comparing the scores of each scale between the leprosarium group and the home group, we found that the home group had higher scores than the leprosarium group in terms of quality of life, psychological domain, physical domain, social support, self-care ability, self-concept, and self-care skills, which was statistically significant (*p* < 0.05). However, we did not find any difference between the two groups in the social domains and the environmental domains, which is shown in Figure 3. Further analyses showed that: the home group was more satisfied with their sex life and the support they received from their friends than the leprosarium group; in the environmental domain, the home group was more confident with the convenience of healthcare services they received and their economic status than the leprosarium group, as shown in Table 3.

The correlation analysis showed that the scores of all domains of social support, all fields of self-care ability, and quality of life were positively correlated (*p* < 0.01), with the strongest correlation occurring between social relationships and social support (0.32, *p* < 0.01). In the home group, all domains of social support, self-care ability, and quality of life scores were positively correlated (*p* < 0.01). Social relationships had the strongest correlation with social support (0.33, *p* < 0.01). In the leprosarium group, the social support score was not significantly correlated with the score of self-care ability and quality of life (*p* > 0.05). However, there was a correlation between the domain portion of all three scales (Figure 4). Self-concept was the highest correlated to support utilization (0.22, *p* < 0.01). 

### 3.3. Structural Estimation Modeling Analysis 

We analyzed the mediating effects of social support with SEM (Figure 5), and the maximum likelihood estimation method was used. The model fit indices showed a good fit: χ2/df = 3.55, CFI = 0.992, TLI = 0.988, RMSEA = 0.043, and SRMR = 0.020 (Table 4). The model coefficients of interest reported in the path diagram were standardized. The direct effect of social support on the quality of life for leprosy-cured patients was 0.33, the direct effect of self-care ability on the quality of life was 0.14, the indirect effect was 0.07, and the direct effect of self-care ability on social support was 0.26.

Next, we performed subgroup analyses of different dwelling types. The model fit indices showed a good fit in the home group: χ2/df = 3.67, CFI = 0.987, TLI = 0.981, RMSEA = 0.046, and SRMR = 0.032. Self-care ability not only directly affects the quality of life but also influences the quality of survival through the mediating role of social support. The direct effect of self-care ability on the quality of life was 0.27, and the mediating effect was 0.08, which corresponds to 22.86%.

In the leprosarium group, the model fit indices showed a good fit: χ2/df = 3.04, CFI = 0.974, TLI = 0.963, RMSEA = 0.060, and SRMR = 0.055. The direct effect of self-care ability on the quality of life was 0.14; no effect of social support on the quality of life or self-care ability on social support was observed (*p* > 0.05). For further information, see Table 5.

Finally, we observed that the domains that had the most significant impact on self-care ability and the quality of life were self-care skills and psychological aspects, both in the leprosarium and home groups. As for social support, different groups performed differently, with objective support having the most significant impact on social support in the home group and subjective support having the most significant impact on social support in the leprosarium group.

## 4. Discussion

Although CLPs have reached the standard of cure, the deformity is often irreversible, which affects their living capacity and which, in turn, seriously affects the quality of life of CLPs. However, there are few studies on the quality of life of CLPs in China. Therefore, we hope that the present study raises the awareness of the government regarding the current situation of life quality of CLPs and further develops effective measures to improve it.

This study investigated 9245 CLPs residing in Jiangsu Province, including 8685 in the home group and 560 in the leprosarium group. The demographic characterization study revealed the following features. Firstly, the ratio of males to females was 2.19:1, and the overall average age was old (74.13 ± 9.03). This is in line with the reported male-to-female ratio of 2.68:1 and the mean age of 69.90 ± 10.65 years in Zhejiang Province [20]. There is no evidence that males are more susceptible to leprosy than females. However, it has been found that males have more access to more intensive social networks, which may be one of the reasons for the more significant number of cured patients among males [21]. It is essential to take comprehensive measures to improve the prevention, control, and treatment of leprosy among males. For example, health education for male patients has been strengthened to enhance their health literacy; more emphasis has been placed on the use of BCG or rifampicin for male close contacts; and stigma and discrimination against male patients should be reduced to increase their willingness to seek medical help. The older age of the CLPs may be because leprosy has been in a low-prevalence state in Jiangsu Province for the past three decades. There are a large number of historical patients [22]. The phenomenon of male-dominated aging leads to a reduction in the labor force, increasing the large expenditures on medical care and pension insurance, burdening the patients’ individuals, families, and society. Additionally, CLPs in Jiangsu Province are mainly farmers, who have a low level of education. Their economic status, living environment, and health resources are generally poor. Studies have found that *M. lepromatosis* can survive for a long time in a specific environment, and farmers are more likely to be infected during long-term labor. Wang et al. [23] concluded that farmers are a high-risk group for leprosy and that there is still deep-rooted discrimination in society against the cured patients and a heavy sense of stigmatization that they suffer. Disease stigmatization exacerbates social discrimination, denies individuals opportunities, and exacerbates social inequalities [24]. Thirdly, the higher unmarried rate in the leprosarium group compared to the home group might be related to the policy of Jiangsu Province. The early segregation policy and social discrimination, fear, and prejudice against leprosy may have contributed to the predominance of unmarried people [25]. Finally, the deformity and social security rate was higher in the leprosarium group than in the home group. At the same time, their living and work capacity were lower [26]. The MDT treatment program can effectively control the incidence of deformity [27]. Care provided by family members can slow down or even prevent the progression of deformity. In addition, as the deformity in the leprosarium group was more severe than that in the home group, their self-care ability and labor ability were also lower.

It can be seen that the life quality and social support of the CLPs in Jiangsu Province were poorer than those of the general population. These results indicate that, despite leprosy being cured, deformities, stigma, and other complications persistently burdened CLPs, especially in the North region of the province [28]. The results indicate that the closer to the North of Jiangsu Province, the worse the quality of life of the CLPs, mainly because of the lack of health resources and underdeveloped economic conditions of the North. The lower economic conditions in the North may expose patients to additional financial pressures. The higher cost of living and relatively few employment opportunities may make it difficult for patients to maintain a good standard of living after recovery. This study found that quality of life, social support, and self-care scores were higher in the home group than in the leprosarium group. For example, the home group was more satisfied with their sexual life and the support they received from friends than the leprosarium group in the social domain [29]. As for the CLPs in leprosaria who found their sexual life was affected, this is mainly due to the impact of leprosy on the marital status of individuals (as CLPs are older and relatively conservative in their thinking, we default to the idea that sexuality is only within marriage). Early isolation policies and social discrimination, fear, and prejudice against leprosy may have contributed to the predominance of unmarried people who survived the leprosarium cure. Experts have also noted that, if a person was confined in a leprosarium, the chances of having sexual relations were reduced. Indeed, if the leprosarium was single sex, the most likely scenario would be homosexual relationships, which, according to cultural values and laws, men might not recognize as occurring. In the environmental domain, the home group was more pleased with the ease of access to healthcare and their economic status than the leprosarium group. The difference in self-care ability was mainly in the self-concept and self-care skills domain, which were higher in the home group than in the leprosarium group. In conclusion, when focusing on a cured patient’s mental health, different treatments should be considered, more flexible healthcare services should be provided, and patients’ self-management skills should be improved.

The results of the correlation analysis show that the quality of life, social support, and self-care ability in the home group were correlated, which was consistent with the results of Chen Wei-ying [30], who analyzed the relationship between these three aspects using a typical correlation. This indicates that social support and self-care ability affect the quality of life. The subgroup analyses showed that the results of the home group were consistent with the results of the total population, while only self-care abilities positively correlated with the quality of life in the leprosarium group. A study by Wu, Li Mei et al. [31] pointed out that improving self-care ability could prevent the occurrence or exacerbation of deformities and complications. Therefore, the confidence of CLPs to return to society was raised, and their quality of life was improved. Differences between the home and leprosarium groups may be due to the leprosarium group’s isolated and closed social life and simple social network. This reality essentially blocks the behavioral and emotional support provided by the family, which seriously jeopardizes the physical and mental health of patients cured of leprosy.

The results of the structural estimation analysis show that both the self-care ability and social support of CLPs could directly affect the quality of life, and self-care ability could also indirectly affect the quality of life through social support. A further subgroup analysis showed that the quality of life in the home group was influenced by self-care, social support, and personality traits to varying degrees. Nevertheless, social support did not correlate with either self-care ability or quality of life in the leprosarium group. This may be due to the isolation and estrangement of the leprosarium group from society, with a simple and closed network of social relations and gradual isolation from the rest of the world. To elucidate this difference more clearly, group discussions were conducted. In the home group, self-care ability can affect the quality of life of CLPs either directly or indirectly by influencing social support. Deps et al. [32] pointed out that good social support can enable cured patients to re-establish values and social relationships and to reintegrate into society more easily after receiving help and support from their family members, friends, and health professionals with relief of stress. Secondly, the mental health of CLPs significantly influenced the quality of life, with a standardized path coefficient of 0.94. Sarode et al. [33] pointed out that, due to the disability caused by the disease, CLPs tend to carry a heavy mental burden. Their mental health is worse than that of the general population [34]. Finally, subjective support had the most significant impact on social support, with a standardized path coefficient of 0.84. This suggests that CLPs place a higher value on being emotionally respected and comforted by others. Good social support is also an important motivation to stimulate self-care, leading cured patients to maintain a relatively good health. In addition to mental health, the leprosarium group differed from the home group in that social relationships significantly impacted the quality of life, with a standardized path coefficient of 0.78. Due to the lack of communication with the outside world, their need for social interaction cannot be met, resulting in a deepening level of need for social relations and social support system. Self-care skills and health literacy mainly influenced self-care ability. This suggests self-care ability can be enhanced, and that the progression of deformity can be effectively curbed by improving the level of knowledge related to disease and care, improving health literacy and self-care skills [35].

In summary, the present study measured the life quality of CLPs in Jiangsu Province, quantifying, for the first time, the burden of the disease on them. To reduce the burden of disease, we suggest the following recommendations: Firstly, CLPs in leprosaria should be actively involved in home care. For those CLPs who cannot return to their families, group-building activities could be organized to strengthen the emotional ties within the leprosarium. Secondly, the community should be mobilized to care for the vulnerable group of CLPs, helping them to enhance their self-confidence and alleviate their negative emotions. Thirdly, the health literacy and self-care capacity of the CLPs should be improved through health education and strengthened training in specialized self-care skills.

## 5. Limitations and Prospects

Firstly, this study was cross-sectional and still has causal inference limitations. Future research should incorporate longitudinal designs to capture dynamic changes and provide a greater understanding of the evolving correlation between self-care ability and social support. Secondly, most participants were elderly, with low literacy. Although a one-to-one interviewer-administered survey was used, information bias still occurred due to the poor recall and misinterpretation of questions. Thirdly, this is a preliminary study that we conducted. In the future, we will conduct further studies to analyze further specific measures to improve the mental health of people cured of leprosy. Finally, the time after finishing leprosy treatment affects CLPs’ self-care ability and quality of life. This study needed to reach a sufficient sample for this outcome. Future research should avoid this potential factor’s impact on the study results.

## 6. Conclusions

The study indicated that the quality of life of CLPs in Jiangsu Province was poorer than that of the general population. The quality of life of the leprosarium group was worse than that of the home group. The quality of life of the cured patients is affected by social support and self-care ability. Social support has a mediating effect between self-care ability and quality of life. Building social support systems and improving the self-care ability of cured leprosy patient could significantly improve their quality of life. More importantly, the stigma attached to leprosy should be reduced.

## Figures and Tables

**Figure 1 healthcare-11-03059-f001:**
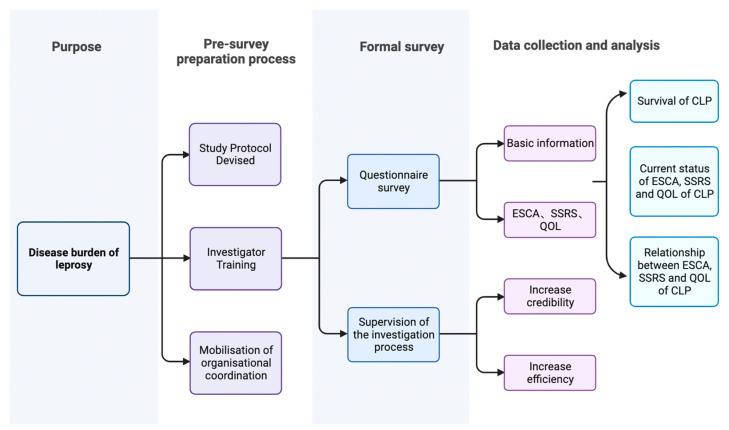
Survey flow chart.

**Figure 2 healthcare-11-03059-f002:**
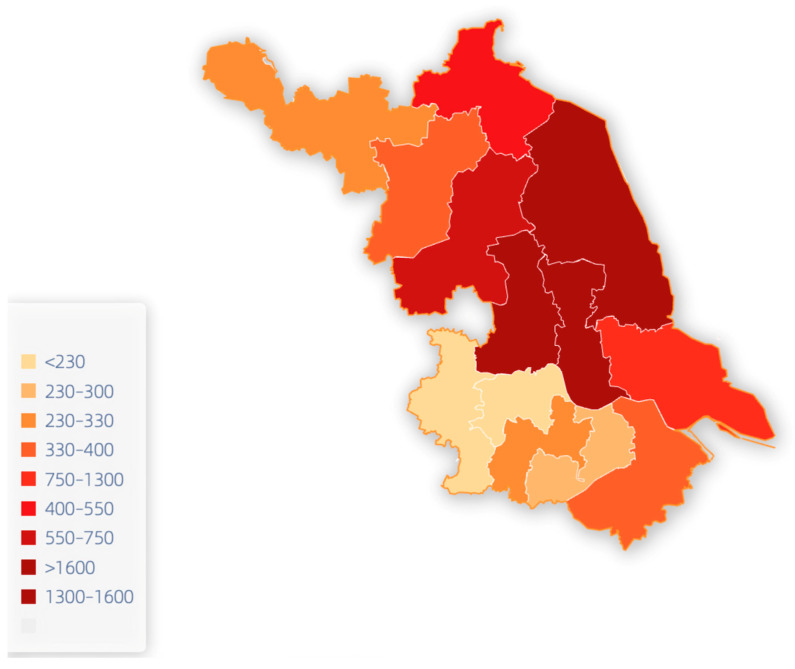
Spatial distribution of the CLPs in Jiangsu Province.

**Figure 3 healthcare-11-03059-f003:**
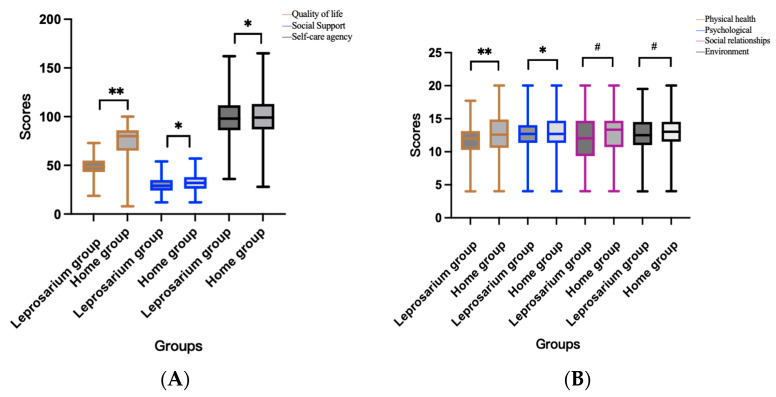
Scores of the leprosarium and home group scales. (**A**) Three scales in different groups; (**B**) four domains of the quality of life for the different groups. * *p* < 0.05; ** *p* < 0.01; # *p* > 0.05.

**Figure 4 healthcare-11-03059-f004:**
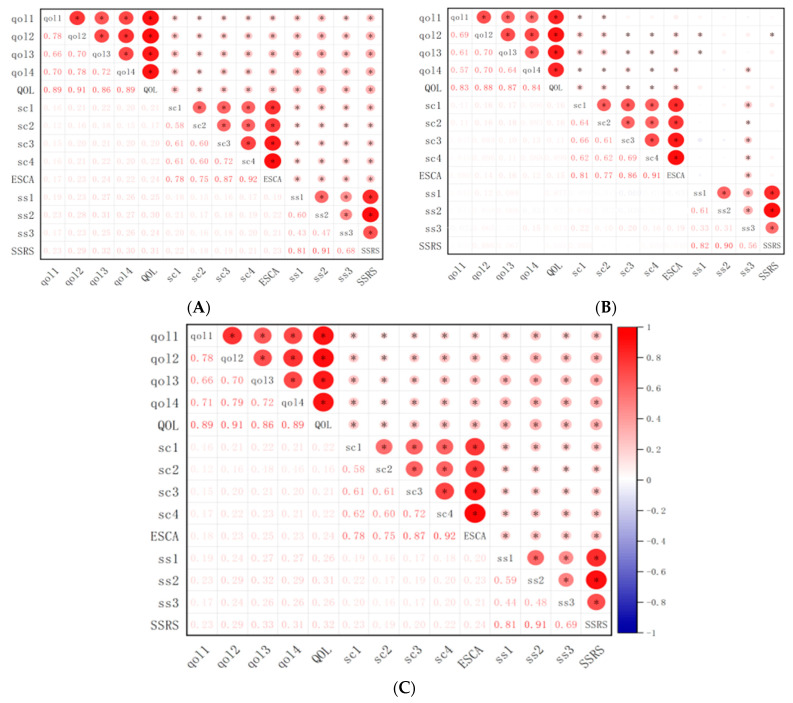
Correlation analysis of social support, self-care ability, and survival quality among the CLPs. (**A**) Total population. (**B**) Leprosarium group. (**C**) Home group. Note: The numbers below are the correlation coefficients, where the redder the color, the higher the correlation coefficient. * means *p* < 0.05. qol1, qol2, qol3, and qol4 are physical health, psychological, social relationships, and environment, respectively; sc1, sc2, sc3, and sc4 is self-concept, self-responsibility, self-care skills, and the level of health literacy, respectively; ss1, ss2, and ss3 are subjective support, objective support, and support utilization, respectively.

**Figure 5 healthcare-11-03059-f005:**
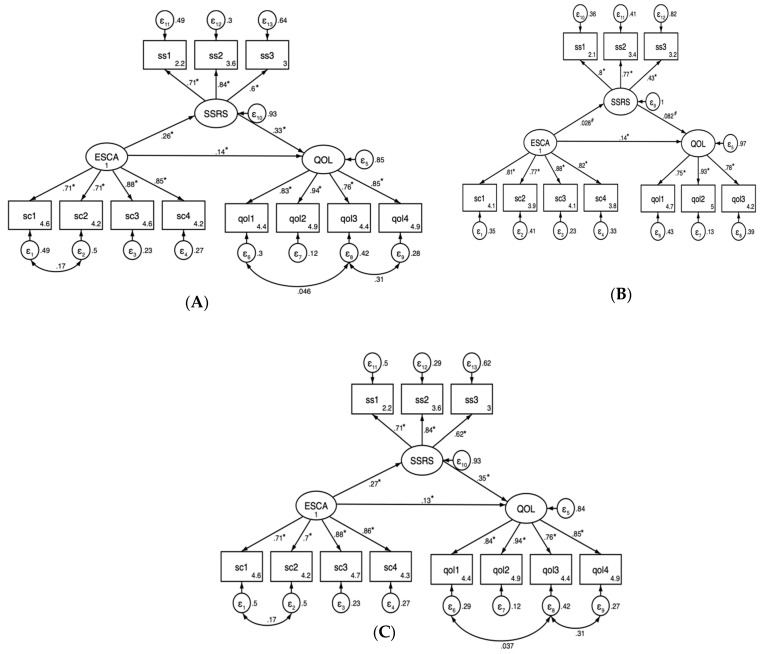
Analysis of self-care ability, social support, and the quality of life pathways of the CLPs. (**A**) The total population. (**B**) Leprosarium group. (**C**) Home group. Note: * *p* < 0.05; # *p* > 0.05. All the coefficients were standardized in the figure. qol1, qol2, qol3, and qol4 are physical health, psychological, social relationships, and environment, respectively; sc1, sc2, sc3, and sc4 are self-concept, self-responsibility, self-care skills, and the level of health literacy, respectively; ss1, ss2, and ss3 are subjective support, objective support, and support utilization, respectively. Age, gender, marriage, occupation, education level, deformity, working ability, and living ability were covariates.

**Table 1 healthcare-11-03059-t001:** Comparison of the basic information of the leprosarium group and the home group.

	Leprosarium Group N (%)	Home Group N (%)	χ^2^	*p*
Sex				
Male	409 (73.04)	5936 (68.35)	5.37	0.02
Female	151 (26.96)	2749 (31.65)		
Age				
<40	1 (0.18)	25 (0.29)	0.58	0.90
40–60	35 (6.25)	490 (5.64)		
60–80	367 (65.54)	5712 (65.77)		
≥80	157 (28.04)	2458 (28.30)		
Marriage				
Single	374 (66.79)	948 (10.92)	916.741	<0.001
Married	119 (21.25)	5687 (65.48)		
Divorced	16 (2.86)	127 (1.46)		
Widowed	50 (8.93)	1907 (21.96)		
Separated	1 (0.18)	16 (0.18)		
Education level				
Illiterate	429 (76.61)	4237 (48.79)	164.767	<0.001
Primary school	108 (19.29)	3334 (38.39)		
High school/above	23 (4.11)	1114 (12.83)		
Occupation				
Farmers	512 (91.43)	8274 (95.27)	23.085	<0.001
Fishermen	0 (0.00)	40 (0.46)		
Other	48 (8.57)	371 (4.27)		
Account				
Agricultural	515 (91.96)	8128 (93.59)	2.275	0.132
Non-agricultural	45 (8.04)	557 (6.41)		
Living capacity				
Fully self-care	75 (13.39)	4970 (57.23)	769.335	<0.001
Partially self-care	290 (51.79)	3184 (36.66)		
Not self-care	195 (34.82)	531 (6.11)		
Work capacity				
Total loss	287 (51.25)	1349 (15.53)	556.49	<0.001
Partial loss	243 (43.39)	3695 (42.54)		
No loss	30 (5.36)	3641 (41.92)		
Deformity				
No	58 (10.36)	4893 (56.34)	447.171	<0.001
Yes	502 (89.64)	3792 (43.66)		
Living expenses				
No	384 (68.57)	7800 (89.81)	233.58	<0.001
Yes	176 (31.43)	885 (10.19)		
Insurance				
No	405 (72.32)	8087 (93.11)	304.018	<0.001
Yes	155 (27.68)	598 (6.89)		

**Table 2 healthcare-11-03059-t002:** Comparison of the quality of life and social support scores between cured leprosy patients and the general population of China.

Domain	CLP (Mean ± SD)	General Population (Mean ± SD)	*t*	*p*
Physical health *	12.69 ± 2.90	15.10 ± 2.30	−27.43	0.000
Psychological *	12.17 ± 2.60	13.89 ± 1.89	−23.56	0.000
Social relationships *	12.85 ± 2.92	13.93 ± 2.06	−13.52	0.000
Environment *	12.86 ± 2.61	12.14 ± 2.08	9.07	0.000
Subjective support #	17.71 ± 4.92	23.34 ± 4.28	−49.61	0.000
Objective support #	7.02 ± 3.18	8.94 ± 2.83	−25.70	0.000
Support utilization #	6.98 ± 2.32	7.99 ± 1.81	−20.54	0.000

Note: Mean ± SD: The mean refers to the average value, while the SD refers to the standard deviation. *: Quality of life among Chinese adults [18]; #: social support among Chinese adults [19].

**Table 3 healthcare-11-03059-t003:** Comparison of the scores of the leprosarium and the home groups regarding the dimensions of the scales.

Variables	Leprosarium Group (Mean ± SD)	Home Group (Mean ± SD)	Statistics	*p*
Quality of life	48.99 ± 9.04	51.39 ± 9.89	9.61	<0.01
Physical health	11.66 ± 2.47	12.76 ± 2.91	29.36	<0.01
1. Pain and discomfort	3.09 ± 0.97	2.69 ± 1.07	48.93	<0.01
2. Dependence on medicinal substances and medical aids	2.96 ± 1.00	2.50 ± 1.08	47.548	<0.01
3. Energy and fatigue	2.75 ± 0.79	2.94 ± 0.91	0.881	0.348
4. Mobility	2.76 ± 0.96	3.08 ± 1.02	0.251	0.617
5. Sleep and rest	3.12 ± 0.86	3.25 ± 0.90	17.926	<0.01
6. Activities of daily living	3.02 ± 0.89	3.19 ± 0.94	28.569	<0.01
7. Work capacity	2.83 ± 0.97	3.04 ± 0.98	0.339	0.56
Psychological	12.44 ± 2.51	12.87 ± 2.61	4.158	0.041
1. Self-esteem	3.01 ± 0.86	3.05 ± 0.85	5.359	0.021
2. Positive feelings	3.03 ± 0.84	3.09 ± 0.80	0.354	0.552
3. Thinking, learning, memory, and concentration	3.01 ± 0.76	3.07 ± 0.81	13.078	<0.01
4. Bodily image and appearance	2.88 ± 0.93	3.13 ± 0.98	4.383	0.036
5. Spirituality/religion/personal beliefs	3.12 ± 0.92	3.26 ± 0.93	11.98	<0.01
6. Negative feelings	2.37 ± 0.94	2.30 ± 0.95	0.001	0.977
Social relationships	12.10 ± 2.87	12.90 ± 2.92	2.767	0.096
1. Personal relationships	3.32 ± 0.85	3.44 ± 0.85	0.493	0.483
2. Sexual activity	2.57 ± 1.01	2.92 ± 0.96	24.093	<0.01
3. Social support	3.21 ± 0.81	3.34 ± 0.83	15.249	<0.01
Environment	12.79 ± 2.62	12.86 ± 2.61	0.358	0.549
1. Freedom, physical safety, and security	3.34 ± 0.84	3.32 ± 0.81	0.627	0.429
2. Home environment	3.34 ± 0.86	3.28 ± 0.81	2.087	0.149
3. Financial resources	2.85 ± 0.86	2.89 ± 0.97	7.082	<0.01
4. Opportunities for acquiring new information and skills	2.91 ± 0.85	2.95 ± 0.88	0.185	0.668
5. Participation in and opportunities for recreation/leisure activity	2.82 ± 0.98	2.83 ± 1.00	1.574	0.21
6. Physical environment (pollution/noise/traffic/climate)	3.56 ± 0.93	3.48 ± 0.87	2.065	0.151
7. Health- and social care: accessibility and quality	3.60 ± 0.86	3.65 ± 0.82	6.856	<0.01
8. Transport	3.18 ± 0.98	3.35 ± 0.93	0.087	0.769
Social Support	29.31 ± 8.02	31.87 ± 8.76	5.81	0.016
Subjective support	16.09 ± 4.68	17.81 ± 4.91	3.115	0.078
Objective support	6.18 ± 3.00	7.07 ± 3.19	3.622	0.057
Support utilization	7.03 ± 2.21	6.98 ± 2.33	1.903	0.168
Self-care agency	98.75 ± 21.90	100.95 ± 19.75	4.577	0.032
Self-concept	18.74 ± 4.62	19.01 ± 4.09	18.995	<0.01
Self-responsibility	13.60 ± 3.45	13.83 ± 3.27	3.334	0.068
Self-care skills	27.47 ± 6.72	27.97 ± 5.97	10.597	<0.01
Level of health literacy	38.94 ± 10.16	40.15 ± 9.41	2.446	0.118

Note: Mean ± SD: The mean refers to the average value, while the SD refers to the standard deviation.

**Table 4 healthcare-11-03059-t004:** Fitting effects before and after the correction of the structural equations.

Group	Index	x^2^/df	SREM	RMSEA	GFI	AGFI	TLI	CFI
	Criteria	<4	<0.1	<0.08	>0.90	>0.90	>0.90	>0.90
Total	Initial Model	33.9	0.029	0.060	0.971	0.954	0.967	0.976
	Modified model	3.55	0.02	0.043	0.987	0.977	0.988	0.992
Leprosarium group	Initial Model	4.52	0.031	0.079	0.944	0.910	0.936	0.952
	Modified model	3.04	0.055	0.060	0.967	0.942	0.963	0.974
Home group	Initial Model	33.54	0.03	0.061	0.969	0.951	0.966	0.975
	Modified model	3.67	0.032	0.046	0.985	0.973	0.981	0.987

Note: SREM: standardized root-mean-squared residual; RMSEA: root-mean-squared error of approximation; GFI: goodness-of-fit index; AGFI: adjusted goodness-of-fit index; TLI: Tucker–Lewis index; and CFI: comparative fit index.

**Table 5 healthcare-11-03059-t005:** Results of the structural equation analysis.

Group	Relationship	β (95%CI)	σ	Z	*p*
Total	ESCA→SSRS	0.26 (0.23, 0.28)	0.01	21.81	<0.01
	ESCA→QOL	0.14 (0.11, 0.16)	0.01	12.16	<0.01
	SSRS→QOL	0.33 (0.31, 0.35)	0.01	29.2	<0.01
	The mediating role of SSRS in ESCA and QOL	0.07			<0.01
Leprosarium group	ESCA→QOL	0.14 (0.04, 0.22)	0.05	2.84	<0.01
Home group	ESCA→SSRS	0.27 (0.24, 0.29)	0.01	22.57	<0.01
	ESCA→QOL	0.13 (0.11, 0.16)	0.01	11.67	<0.01
	SSRS→QOL	0.35 (0.32, 0.37)	0.01	32.07	<0.01
	The mediating role of SSRS in ESCA and QOL	0.08			<0.01

Note: ESCA: Self-care Ability; SSRS: Social Support; QOL: Quality of Life; β represents standardized coefficients; and σ represents the standard error.

## Data Availability

The data analyzed in this study are subject to the following licenses/restrictions. Data supporting the results of this study are available from the Jiangsu Provincial Center for Disease Control and Prevention; however, the availability of these data is limited, they should be used with permission from this study, and they are not publicly available. These data are available to the authors upon reasonable request and with permission from the Jiangsu Provincial Center for Disease Control and Prevention. Requests to access these datasets should be directed to zhanglh@jscdc.cn.

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
