# Peer review of "Self-Care Ability and Life Quality of Cured Leprosy Patients: The Mediating Effects of Social Support"

_healthcare, 2023, doi:10.3390/healthcare11233059_

Round 1
Reviewer 1 Report
Comments and Suggestions for Authors
Dear authors,
Please see below my comments:
1) Please explain why questionnaires have been chosen as instruments of data collection. Have you consider online questionnaires or interviews?
2) Interesting data regarding the male to female ratio, which is well explained as seen below:
Firstly, the ratio of males to females 320 was 2.19:1, and the overall average age was older (74.13 ± 9.03). in line with the reported 321 male-to-female ratio was 2.68:1, and the mean age was 69.90 ± 10.65 years in Zhejiang 322 Province and elsewhere [32]. There is no evidence that males are more susceptible to lep- 323 rosy than females'
However a paragraph in future work is missing, explaining how this ratio can be reduced.
Comments on the Quality of English LanguageDear authors,
Some minor corrections and editing are required in introduction, discussion and conclusion.
Author Response
Dear reviewer,
Thank you for your comments concerning our manuscriptentitled "Associations of social support with self-care ability and quality of life in leprosy cured patients: A structural equation-based modeling study”(Manuscript ID: healthcare-2682600). Those comments are valuable and very helpful. We have read through comments carefully and have made corrections. Based on the instructions provided in your letter, we uploaded the fle of the revised manuscript. The following document is our point by point response to your comments.
We're very sorry, but we made final changes to the manuscript so the number of lines changed, so we've re-uploaded the new file called Revised-Reply1.
We would love to thank you for allowing us to resubmit a revised copy of the manuscript and we highly appreciate your time and consideration.
Kind regards
LiXu

Reviewer 2 Report
Comments and Suggestions for Authors
The paper is well written and presented.
The title could be better phrased and made shorter.
It would be helpful, if the abstract were made shorter and discuss the most relevant and important results and conclusions.
Please use the appropriate format when mentioning names of bacteria: M.lepromatosis (should be italics)
Author Response
Dear reviewer,
Thank you for your comments concerning our manuscriptentitled "Associations of social support with self-care ability and quality of life in leprosy cured patients: A structural equation-based modeling study”(Manuscript ID: healthcare-2682600). Those comments are valuable and very helpful. We have read through comments carefully and have made corrections. Based on the instructions provided in your letter, we uploaded the fle of the revised manuscript. The following document is our point by point response to your comments.
We're very sorry, but we made final changes to the manuscript so the number of lines changed, so we've re-uploaded the new file called Revised-Reply2.
We would love to thank you for allowing us to resubmit a revised copy of the manuscript and we highly appreciate your time and consideration.
Kind regards
LiXu

Reviewer 3 Report
Comments and Suggestions for Authors
The authors present a study on quality of life of an impressive number of patients cured from leprosy.
1. The Abstract is much too long. Is this in accord with the journal's instructions? If not, please reduce. There are repetitions that are not necessary.
2. Methods: The authors do not adequately describe their patients for the international reader. What does a leprosarium consist of? How is it decided that a patient will be treated at home or leprosarium? Does the patient's financial situation, family conditions or other, influence where the treatment will be performed? In this case comparing the 2 groups will be biased since they are selected differently.
The recruitement process is not described either.
Ethical considerations are not mentioned.
In the flowchart: What do you mean by developing a program? Program for what? And what is meant/done under Monitoring and feedback?
All those points should be made completely clear when reading the Methods.
3. Results: Table 1 does not indicate what uniot is used. I suppose 'Number'. There is a large difference between the number of participants between the 2 groups. Numbers become meaningless and are typically alse given as %. The table gives little meaning the way it is presented., and of course p will be very low if only total numbers, nopt % are calculated. Table 2 does not explain what units are used either (mean +-SD)? other? Thgis is not well explained in the text either (ome places mean is given, but not the numbers that follow).
4. How is cure defined? You mention progression of deformities. Can those individuals be defined as cured?
5. Your aim: 'It aims to improve the quality of survival of cured leprosy patients, suggestions for the development of strategies to reduce the leprosy disease burden in Jiangsu Province, as well as in similar areas of the country and the world.'
The design of your study does not use any methods that would improve quality of life, neither do you explore strategies to reduce burden, and cannot generalize for the world. Your study only measures the burden, not means that may improve ql, reduce burden, etc.
6. The paper could be shortened, there are some repetitions, e.g. the instruments described both in the introduction and Methods.
7. The Conclusion is much too long. Abstract and Conclusion should be brief with the purpose of giving the reader a quick overview of the study and the concluded results.
Comments on the Quality of English Language
English is mostly fine, but some sentences are structured in a way that makes the somewhat difficult to understand.
Author Response

(The authors gave the same response as above.)

Reviewer 4 Report
Comments and Suggestions for Authors
Thanks for this study. Research on the late effects of leprosy is very relevant- as it highlights the importance of prevention, early diagnosis and treatment. This is particularly true in a highly endemic region such as the Jiangsu region.
I read your manuscript with great interest, but I feel the text would benefit from a thorough revision. In general, I want to mention that the sentences are very long- which makes it hard for the interested reader. This is particularly true for the abstract and introduction, which should be reduced in size (eg. lines 12-17, 62-66).
Furthermore, in your introduction, could you clarify to which population these gains in DALYs apply in lines 72-76)?
Was this research approved by an ethical committee or institutional review board?
Could you disclose statistics on the time after finishing leprosy treatment and the surveys? That would be helpful in interpreting the results.
Comments on the Quality of English LanguageThe English is definitely correct from a grammar point of view. However, sentences are very long and therefore hard to read. That is particularly true for the abstract and introduction section. Please revise these- it should
Author Response
Dear reviewer,
Thank you for your comments concerning our manuscriptentitled "Associations of social support with self-care ability and quality of life in leprosy cured patients: A structural equation-based modeling study”(Manuscript ID: healthcare-2682600). Those comments are valuable and very helpful. We have read through comments carefully and have made corrections. Based on the instructions provided in your letter, we uploaded the fle of the revised manuscript. The following document is our point by point response to your comments.
We're very sorry, but we made final changes to the manuscript so the number of lines changed, so we've re-uploaded the new file called Revised-Reply4.
We would love to thank you for allowing us to resubmit a revised copy of the manuscript and we highly appreciate your time and consideration.
Kind regards
LiXu

Reviewer 5 Report
Comments and Suggestions for Authors
In the manuscript titled "Associations of Social Support with Self-Care Ability and 2 Quality of Life in Leprosy Cured Patients: A Structural Equa- 3 tion-Based Modeling Study", authors explore and understand various aspects of the disease burden, social support, self-care ability, and quality of life among individuals cured of leprosy in Jiangsu province. Please see my suggestions below:
1. Please consider re-writing the abstract. The first paragraph is a very long one statement, please consider to break it into 2-3 sentences.
2. Minor grammatical errors occur throughout the manuscript, please proofread for those errors.
3. I understand authors talk about the disease burden and the authors are trying to compare home vs lep groups but I am still confused about the purpose of the study and how it is significant. Please mention and elaborate what is the goal of the study and its significance.
4. What were the controls in this study?
5. Pearson correlation is used when the data is normally distributed. But it is not, can the authors comment why they used Pearson and not Spearman correlation?
6. Please enlarge figure 4
7. Please state limitations of the study
Comments on the Quality of English Language1. In line 314, word "provinces" is of different font
Author Response
Dear reviewer,
Thank you for your comments concerning our manuscriptentitled "Associations of social support with self-care ability and quality of life in leprosy cured patients: A structural equation-based modeling study”(Manuscript ID: healthcare-2682600). Those comments are valuable and very helpful. We have read through comments carefully and have made corrections. Based on the instructions provided in your letter, we uploaded the fle of the revised manuscript. The following document is our point by point response to your comments.
We're very sorry, but we made final changes to the manuscript so the number of lines changed, so we've re-uploaded the new file called Revised-Reply5.
We would love to thank you for allowing us to resubmit a revised copy of the manuscript and we highly appreciate your time and consideration.
Kind regards
LiXu

Round 2
Reviewer 3 Report
Comments and Suggestions for Authors
Some comments are addressed, but others are not adequately so.
1. You describe the recruitment process in future tense. I assume this is a process that has already happened and should be in past tense. Point 4 in the recruitment: I do not understand this sentence. 'The contents mainly include the purpose of the study, confidentiality terms and measures, Etc.' Please reformulate, it does not seem to be written as a point in a manner used for a medical journal.
2. When replying to the question in the flow chart you do not mention any changes in the text. It is not sufficient that this reviewer is given an explanation, it should be given to all readers of the paper.
3. Tables 1, 2 and 3 are not changed adequately as stated in the point-by-point response. Please do not claim having made adequate changes when this is not done. You still do not explain the units used, and do not give % where necessary.
4. You still do not explain how cured patients would develop deformity progression. What do you mean? Were they not completely cured?
5. You can not claim that your study aims to improve ql. This point is not addressed. Please see Q5 again.
Author Response
Dear reviewer:
We are sorry that we did not effectively solve the problems you raised in the first revision, so as to make clearer revisions to the manuscript. Thanks for your help with our manuscript, and we are impressed with the professional comments. We have responded to those comments point by point, with the response sections in italics and purple. We have highlight all the amends on the manuscript or indicate them by using tracked changes.
Kind Regards,
LiXu

Reviewer 4 Report
Comments and Suggestions for Authors
Dear authors,
Thanks for this revision and for your clarifications. My main initial concerns were pertaining the presentation of the text, and in the revision this has greatly improved in terms of length, flow and readability. Thanks for that! I do not have any further suggestions.
Comments on the Quality of English Language-
Author Response
Dear reviewer,
Thank you very much for taking time out of your busy schedule to review our manuscript and give us your valuable feedback.
We are very pleased to receive your response. We would like to thank you for your hard work and professional input. If you have any additional questions or need further information in the future, please feel free to contact us.
Thank you again for your support and assistance!
Kind Regards,
LiXu